# Dynamic Rescaling for Training GNNs

**Nimrah Mustafa**
CISPA
66123 Saarbrücken, Germany
nimrah.mustafa@cispa.de

**Rebekka Burkholz**
CISPA
66123 Saarbrücken, Germany
burkholz@cispa.de

## Abstract

Graph neural networks (GNNs) with a rescale invariance, such as GATs, can be re-parameterized during optimization through dynamic rescaling of network parameters and gradients while keeping the loss invariant. In this work, we explore dynamic rescaling as a tool to influence GNN training dynamics in two key ways: i) balancing the network with respect to various criteria, and ii) controlling the relative learning speeds of different layers. We gain novel insights, unique to GNNs, that reveal distinct training modes for different tasks. For heterophilic graphs, achieving balance based on relative gradients leads to faster training and better generalization. In contrast, homophilic graphs benefit from delaying the learning of later layers. Additionally, we show that training in balance supports larger learning rates, which can improve generalization. Moreover, controlling layer-wise training speeds is linked to grokking-like phenomena, which may be of independent interest.

Deep neural networks (DNNs) with positively homogeneous non-linear activation functions such as ReLU exhibit the rescale invariance property [31], i.e. scaling down incoming weights to a neuron and scaling up the outgoing weights from the neuron by the same factor does not alter the function represented by the network.

Under gradient flow that assumes an infinitesimally small learning rate, this rescale symmetry induces the conservation of a relationship between network parameters and gradients, determined by the initial state [8]. More specifically, it is known that for traditional DNNs and convolutional neural networks (CNNs) with homogenous activation functions such as ReLUs, the difference between the squared $L2$-norms of incoming and outgoing parameters to a neuron stays constant (and is thus conserved). When this conserved quantity is (nearly) zero, the network is said to be in a balanced state. Training DNNs is generally considered well-conditioned by a balanced state of this conservation law.

In the context of Graph Neural Networks (GNNs), the first work presenting insights regarding norm balance derives the conservation law for GATs [2] and demonstrates how a balanced initialization enhances the trainability of GATs, particularly deeper networks [28]. This balanced state that would hold under gradient flow could potentially be beneficial throughout training. However, in practice, factors that drive optimization such as finite learning rates, momentum, weight decay, and batch stochasticity break the rescale (and other) symmetries [15] and consequent conserved quantities, causing the network to topple out of balance due to which training conditions may deteriorate.

Motivated by the positive outcomes of training a model balanced at initialization, we utilize the identified rescale invariance of GATs [28] to further investigate the effects of maintaining this balance throughout training rather than only at initialization, by *dynamic rescaling*, i.e. scaling network parameters at the neuron level during training in a loss-invariant manner. To this end, we derive a general procedure to balance a GAT network, not necessarily only w.r.t. parameter weight norms, but any criterion that is a function of network parameters and gradients. More specifically, we propose a novel criterion based on relative gradients of network parameters and demonstrate that this criterion, or a combination of both criteria, often offers practical gains in terms of training speedup or better generalization.

38th Conference on Neural Information Processing Systems (NeurIPS 2024).

The implications of the core concept of dynamic rescaling extends beyond training in balance. For example, it enables arbitrary control of the order in which network layers learn during training. This can be viewed as inducing an imbalanced state in the network that may be desirable in some cases [16]. Based on the experimental exploration of our methodological ideas, we draw various novel insights into how dynamic rescaling can potentially be leveraged. Firstly, specifically regarding graph learning, we discover potential trends of optimal learning dynamics for homophily and heterophily. We observe that, in terms of training speed and generalization, training all network layers in balance tends to be more beneficial for heterophilic than homophilic graphs. On the contrary, homophilic graphs tend to benefit from more focused learning in the first (earlier) layers. Secondly, we hypothesize that larger learning rates that tend to increasingly disrupt network balance can be better supported by dynamic rebalancing of the network in improving generalization. Thirdly, we encounter other interesting phenomena similar to grokking [33] that we relate to our insight on layer learning order.

This work is also motivated by earlier studies on deep feedforward non-linear neural networks that exploit rescale invariance, using transformations respecting loss-invariant symmetry to teleport parameters to another point in the loss landscape with steeper gradients to improve optimization [46] and/or convergence rate guarantees [47]. To the best of our knowledge, we are the first to conduct an initial exploration of these ideas on GNNs, where identifying and exploiting the rescale invariance is not as straightforward, due to their peculiar architectural design elements such as node-wise neighborhood aggregation that make identifying the corresponding rescale invariances challenging. Thus, this largely remains unexplored territory for GNNs and our understanding of the underlying gradient dynamics in GNNs lags in comparison to DNNs. Rather than proposing a one-size-fits-all solution or achieving state-of-the-art performance, this paper aims to contribute insights into GNN learning dynamics.

In summary, our contributions are as follows:

1. We prove that, given a GNN exhibits a rescale invariance (like GCNs or GATs), we can manipulate gradient norms while leaving the GNN and the loss invariant and thus influence the learning dynamics during optimization.

2. We derive the procedure to balance a GAT network w.r.t. any criterion that is a function of network parameters and gradients by dynamic rescaling.

3. We suggest a novel criterion for balancing based on relative gradients and find it to be promising for improving generalization and training speed in practical settings.

4. We explore our conceptual ideas empirically and find promising directions to utilize dynamic rescaling for more practical benefits, by training in balance or controlling order of learning among network layers.

5. We discuss novel insights regarding i) trends in training dynamics for homophilic and heterophilic graphs ii) larger learning rates and iii) interesting grokking-like phenomena.

# 1 Related work

There has recently been an increased interest in studying training dynamics[43] and generalization [39]. Approaches to address trainability issues in GNNs include initialization [20, 28, 13, 12], normalization [3, 5, 50, 48], skip-connections[13], regularization[34, 44], their combinations[25, 4], architectural variations[29], and insights from graph signal processing based on spectral properties [45]. While our approach of dynamic rescaling allows us to use gradients in the network to control the rate at which GNN layers (and potentially neurons) learn non-linear transformations of their features,[35] draws an interesting parallel by modulating message passing updates based on gradients to control the rate at which nodes learn. The closest work to ours is [28] which proposes using balanced norms at initialization. Our work differs in mainly three ways as we: i) derive how a balanced state can be achieved not only at initialization but also during training by dynamic rescaling, ii) propose a different criterion for balance based on relative gradients, and iii) present insights on training in and out of balance in light of input graph homophily and heterophily.

For traditional DNNs, there is a deeper understanding of loss invariant symmetries and their impact on gradient dynamics[8, 17, 40]. Several studies exploit rescale symmetry and corresponding conservation laws in (feed-forward and convolutional) neural networks in various ways to aid optimization [31, 26, 46], regularization [37], and compression[38]. [47] introduce a set of nonlinear,

data-dependent symmetries, relate conserved quantities to the convergence rate and sharpness of the optima, and provide insights into how initialization impacts convergence and generalizability. Gradient flow equations for neural networks have also been extended to account for realistic optimization elements such as finite learning rates, momentum, weight decay, etc. [15].

## 2 Dynamic rescaling

**Preliminaries** Consider a $L$ layer GAT network $f$ with positively homogeneous activation $\phi$ (i.e $\phi(x) = x\phi'(x)$) and consequently, $\phi(ax) = a\phi(x)$ for positive scalars $a$) such as ReLU $\phi(x) = \max\{x, 0\}$ or LeakyReLU $\phi(x) = \max\{x, 0\} + -\alpha \max\{-x, 0\}$. Then, as shown by [28], the parameters $\mathbf{W}^l[i,:]$, $\mathbf{W}^{l+1}[:,i]$, and $\mathbf{a}^l[i]$ associated with a hidden unit $i$ in the network layer $l$, may be respectively scaled to $\tilde{\mathbf{W}}^l[i,:] = \lambda\mathbf{W}^l[i,:]$, $\tilde{\mathbf{W}}^{l+1}[:,i] = \lambda^{-1}\mathbf{W}^{l+1}[:,i]$ and $\tilde{\mathbf{a}}^l[i] = \lambda^{-1}\mathbf{a}^l[i]$ where $\lambda > 0$ such that $f = \tilde{f}$, i.e. the rescaling respects the network symmetry.

Note that $\mathbf{W}^l[i,:]$ and $\mathbf{W}^{l+1}[:,i]$ denote weights incoming to and outgoing from neuron $i \in [d_l]$, respectively, where $d_l$ is the width of layer $l$. Given a network parameter $\theta$ and its gradient $\nabla_\theta\mathcal{L}$ w.r.t. the network loss $\mathcal{L}$, the relative gradient $\Delta\theta$ of parameter $\theta$ is defined as:

$$\Delta\theta = \nabla_\theta\mathcal{L}/\theta \ \text{ for } \ \theta \neq 0 \ \text{ or } \ \Delta\theta = 0 \ \text{ for } \ \theta = 0. \tag{1}$$

This rescale property is more powerful than it might appear at first, as it provides us the means to significantly influence the training dynamics. It suggests that we have a high number of degrees of freedom to pick a parameterization without changing the function of a GAT. Concretely, we can use any scaling factors $\lambda_i^{(l)} > 0$ that are associated with features in the middle layers and define another parameterization $\tilde{\mathbf{W}}^l[i,j] = \lambda_i^{(l)}/\lambda_j^{(l-1)}\mathbf{W}^l[i,j]$ and $\tilde{\mathbf{a}}^l[i] = \mathbf{a}^l[i]/\lambda_i^{(l-1)}$ that will induce the same function. Yet, according to the following lemma, the gradients of the parameters depend on the rescale factors and can therefore be controlled correspondingly.

**Lemma 2.1** (Gradient scaling). *Under the rescale invariance of GATs, if a parameter is scaled by $\tilde{\theta} = \lambda\theta$, then its gradient is scaled as $\nabla_{\tilde{\theta}}\mathcal{L} = \lambda^{-1}\nabla_\theta\mathcal{L}$.*

We defer the proof to the appendix A.1. It generally implies that we have the freedom to pick any positively homogeneous constants so that our resulting gradients $\nabla_{\tilde{\mathbf{W}}^l[i,j]}\mathcal{L} = \lambda_j^{(l-1)}/\lambda_i^{(l)}\nabla_{\mathbf{W}^l[i,j]}\mathcal{L}$ and $\nabla_{\tilde{\mathbf{a}}^l[i]}\mathcal{L} = \lambda_i^{(l-1)}\nabla_{\mathbf{a}^l[i]}\mathcal{L}$ induce favorable learning dynamics. Considering the flexibility, in which gradient direction we can move during gradient descent, the choice likely has a significant influence on our learning success. We aim to exploit this fact in this work and discover conceptual insights into what criteria could constitute choices. Thus, we follow the following procedure.

**Balancing criteria and procedure** The rescale invariance property allows us to rescale parameters to fulfill the desired criterion not only at initialization but also during training without changing the network output while potentially improving the training dynamics.

We could indeed choose a relatively general criterion $g : \mathbf{R^m} \to \mathbf{R^m}$ that depends on our rescaled parameters and gradients and determines our choice of scaling factors. As long as we can solve

$$g\left(\lambda_i^{(l)}\mathbf{W}^l[i,:], \lambda_i^{(l)-1}\mathbf{W}^{l+1}[:,i], \lambda_i^{(l)-1}\mathbf{a}^l[i], \lambda_i^{(l)-1}\nabla_{\mathbf{W}^l[i,:]}\mathcal{L}, \lambda_i^{(l)}\nabla_{\mathbf{W}^{l+1}[:,i]}\mathcal{L}, \lambda_i^{(l)}\nabla_{\mathbf{a}^l[i]}\mathcal{L}\right) = 0. \tag{2}$$

, For unique nonzero scaling parameters, $g$ can act as our guide during gradient descent. Importantly, not all reasonable criteria determine the scaling factors.

As has been derived recently for GATs [28], the rescale invariance also induces a conservation law that holds throughout training and is characterized by the fact that the scaling factors cancel out. In fact, any such law that remains invariant under specific parameter transformations is also linked to an invariance like the rescale invariance. according to Noether's theorem. Specifically, in the case of GATs, the following equations

$$\langle W^l[i,:], \nabla_{W^l[i,:]}\mathcal{L}\rangle - \langle a^l[i], \nabla_{a^l[i]}\mathcal{L}\rangle - \langle W^{l+1}[:,i], \nabla_{W^{l+1}[:,i]}\mathcal{L}\rangle = 0. \tag{3}$$

hold regardless of the scaling factors. This law implies that a specific sum of L2-norms of the corresponding parameters stays conserved throughout gradient descent if the learning rates are sufficiently small.

Recently, it has been shown that balancing these squared parameter $l2-$norms at initialization such that $\left\|\mathbf{W}^l[i,:]\right\|^2 - \left\|\mathbf{a}^l[i]\right\|^2 - \left\|\mathbf{W}^{l+1}[:i]\right\|^2 = 0, \forall i \in d_l, l \in [L-1]$, induces good initial trainability in GATs [28]. For larger learning rates, this balance might get disturbed during training. Yet, we could use our rescaling degrees of freedom to bring the parameters back in balance. To induce even better trainability, we propose another criterion to rescale a GAT network, which balances the norms of gradients relative to the corresponding parameters, i.e., $\Delta\theta = \nabla_\theta \mathcal{L}/\theta$. Intuitively, this should allow the parameters in different layers to move at similar speeds and ensure good trainability in all parts of the network.

$$\left\|\Delta\mathbf{W}^l[i,:]\right\|^2 - \left\|\Delta\mathbf{a}^l[i]\right\|^2 - \left\|\Delta\mathbf{W}^{l+1}[:i]\right\|^2 = 0. \tag{4}$$

Yet, as this criterion is not naturally preserved during gradient descent or gradient flow like the weight norms, fulfilling the equation above requires frequent rescaling during training. Thus, based on Eq.(2), balancing a neuron $i$ in layer $l$ w.r.t. relative gradients requires fulfilling:

$$\left\|\frac{\lambda_i^{(l)-1}\nabla_{\mathbf{W}^l[i,:]}\mathcal{L}}{\lambda_i^{(l)}\mathbf{W}^l[i,:]}\right\|^2 - \left\|\frac{\lambda_i^{(l)}\nabla_{\mathbf{W}^{l+1}[:,i]}\mathcal{L}}{\lambda_i^{(l)-1}\mathbf{W}^{l+1}[:,i]}\right\|^2 - \left\|\frac{\lambda_i^{(l)}\nabla_{\mathbf{a}^l[i]}\mathcal{L}}{\lambda_i^{(l)-1}\mathbf{a}^l[i]}\right\|^2 = 0. \tag{5}$$

Balancing the entire network requires fulfilling Eq.(5) $\forall i \in [d_l], l \in [L-1]$. In this case, note that every weight $\mathbf{W}^l[i,j]$ is eventually scaled by $\lambda_i^{(l)}/\lambda_j^{(l)}$. Thus, balancing the entire network requires iterative rescaling until convergence of all rescaling factors to 1.

For each iteration $t \in [1,T]$ and given $\lambda_i^{(l)^{(0)}} = 1, \forall i \in [d_l], l \in [L-1]$, the scaling factor $\lambda_i^{(l)^{(t)}}$ is given by:

$$\lambda_i^{(l)^{(t)}} = \left(\frac{\left\|\Delta\mathbf{W}^l[i,:]^{(t-1)}\right\|^2}{\left\|\Delta\mathbf{a}^l[i]^{(t-1)}\right\|^2 + \left\|\Delta\mathbf{W}^{l+1}[:i]^{(t-1)}\right\|^2}\right)^{\frac{1}{8}} \quad ; \text{ where} \tag{6}$$

$$\Delta\theta^{(t)} = \begin{cases} \frac{\lambda_i^{(l)(t)-1}\nabla_\theta\mathcal{L}}{\lambda_i^{(l)(t)}\theta} & \text{if } \theta \in \mathbf{W}^l[i,:]^{(t-1)} ; t > 0 \\ \frac{\lambda_i^{(l)(t)}\nabla_\theta\mathcal{L}}{\lambda_i^{(l)(t)-1}\theta} & \text{if } \theta \in \{\mathbf{W}^{l+1}[:,i]^{(t-1)}, \mathbf{a}^l[i]^{(t-1)}\} ; t > 0. \end{cases} \tag{7}$$

Ideally, this process is repeated to convergence until $\lambda_i^{(l)^{(T)}} = 1, \forall i \in [d_l], l \in [L-1]$. The number of required iterations depends on the frequency of rebalancing during training as well as the network parameters and gradients. In practice, we find that this is not a too computationally expensive process and a few iterations $(< 10)$ are sufficient to (mostly if not completely) balance the network. In terms of time complexity, this only affects the training time linearly depending on the frequency of rebalancing, which is a controllable hyperparameter.

**Implications**   Dynamic rescaling opens up the opportunity to control training dynamics in several ways for which we lay out two key ideas as follows.

Firstly, dynamic rescaling can be used to train networks in balance w.r.t. certain criteria. We propose that balancing based on relative gradients may be one such good candidate. Our intuition is that balanced relative gradients allow all layers (and neurons) in the network a relatively equal opportunity to learn by propagating gradients to drive parameter change throughout the network, thereby enhancing trainability. We observe that this novel insight shows the promising potential of being translated into practical gains such as faster or better generalization on real-world data, particularly for heterophilic tasks.

Secondly, dynamic rescaling allows us to control of relative training speed at the level or neuron level. In principle, by rescaling layers, we can configure the relative order in which they learn arbitrarily, at any time during training. This is a direct consequence of the conservation law that parameters and their gradients in the network adhere to. The underlying insight is that by controlling the parameter weight or relative gradient norms, the gradients that drive parameter change can be influenced. As a result, a layer receiving relatively larger gradients (due to relatively scaled-down

Table 1: Results of training a 5-layer GAT network with various dynamic rescaling (DR) settings. The mean $\pm 95\%$ CI test metric at the epoch of the best validation metric across 10 splits is reported using the best learning rate from $\{0.01, 0.001, 0.005\}$. The evaluation metric is accuracy for `roman-empire` and `amazon-ratings`, and ROC AUC for the remaining three datasets.

|           | roman-empire | amazon-ratings | tolokers | questions | minesweeper |
|-----------|:---:|:---:|:---:|:---:|:---:|
| w/o DR    | $.4978 \pm .0209$ | $.4545 \pm .0043$ | $.6493 \pm .008$ | $.5829 \pm .0172$ | $.5057 \pm .0058$ |
| $DR_W$    | $.3307 \pm .0670$ | $\mathbf{.4547 \pm .0041}$ | $.6451 \pm .0135$ | $.5791 \pm .0145$ | $.5058 \pm .0051$ |
| $DR_{RG}$ | $.5422 \pm .0234$ | $.4540 \pm .0029$ | $.6637 \pm .0088$ | $\mathbf{.5869 \pm .0145}$ | $.5065 \pm .0076$ |
| $DR_C$    | $\mathbf{.6731 \pm .0149^{*}}$ | $.4526 \pm .0042$ | $\mathbf{.6642 \pm .0064}$ | $.5696 \pm .0132$ | $\mathbf{.5080 \pm .0053}$ |

weight norms) thus learns 'more' or 'faster' than other layers. While the possibilities are numerous, in this work, we limit our investigation to a simple but core case: allowing the network to (initially) focus learning on a specific layer by scaling down its weight norm at initialization. We observe that initially allowing more focused learning in the earlier layers of the network can improve the convergence time substantially while retaining or even improving the generalization for homophilic tasks.

Our findings suggest that the order in which layers learn influences convergence time and generalization. This building block may be leveraged to devise more sophisticated learning sequences such as training layers cyclically or in a task-dependent manner rather than in a predefined order. We elaborate on the current limitations of dynamic rescaling in the appendix.

While these concepts apply to any GNN, provided its rescale invariance has been identified, we focus our investigation on the GAT architecture in this work as they are a generalization of the more basic GCN architecture that is the building block of more complex GNNs. Furthermore, GAT serves as a strong basis for graph learning with an attention mechanism, in which there has been an increased interest recently [18, 9].

## 3 Experiments

We divide our exploration of the ideas discussed in §2 into three parts: 1) practical gains on real-world data of training in balance by dynamic rescaling, 2) empirical insights into the layer-level order of learning, and 3) observation of a grokking-like phenomenon, which is related to 2). Hereafter, we use the notation $DR_W$, and $DR_{RG}$ to denote dynamic rescaling w.r.t. weight norms and relative gradients, respectively, every $10^{th}$ epoch. $DR_C$ denotes a combination of the two by rescaling w.r.t. weight norms every $10^{th}$ epoch and w.r.t. relative gradients in all other epochs. A maximum of 10 iterations for the rebalancing procedure outlined in Eq. (6) and (7) were used. All experiments use the Adam optimizer and networks are randomly initialized with looks-linear orthogonal structure [36, 1] unless specified otherwise. Experiments were run on an NVIDIA RTX A6000 GPU with 50GB RAM. Our experimental code is available at `https://github.com/RelationalML/Dynamic_Rescaling_GAT`.

### 3.1 Training in balance

We primarily study the effect of training GAT in a balanced state based on the relative gradients criterion (see Eq.(4)), by dynamic rescaling on five real-world heterophilic benchmark datasets [32].

We find that rebalancing the network w.r.t. relative gradient norms is more effective than the criterion based on parameter weight norms, as shown in Table 1. This aligns with observations on CNNs where rebalancing w.r.t. parameter norms was also not effective [37]. Therefore, our insight on the impact of rebalancing w.r.t. relative gradients may also be of independent interest outside the context of GNNs.

In addition to improved generalization in most cases, dynamic rescaling may also provide the benefit of fewer training epochs to attain comparable or even slightly better generalization, as shown in Fig. 1. We make an interesting observation that a balanced state during training together with larger learning rates results in better generalization than when either of the two components is individually employed.

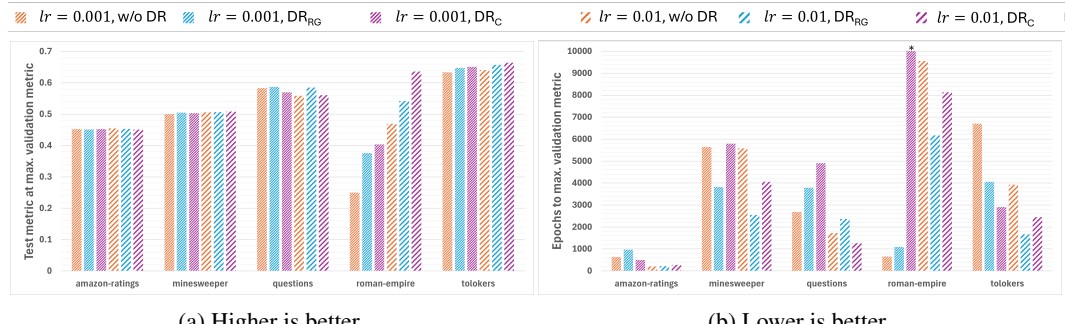

(a) Higher is better.  (b) Lower is better.

Figure 1: Performance of a 5-layer GAT with various dynamic rescaling (DR) settings using learning rates (lr) $0.001$ and $0.01$. Across 10 splits, the mean accuracy is reported for `roman-empire` and `amazon-ratings` while ROC AUC is reported for `minesweeper`, `questions` and `tolokers`. The case annotated by * indicates training for more than 10k epochs.

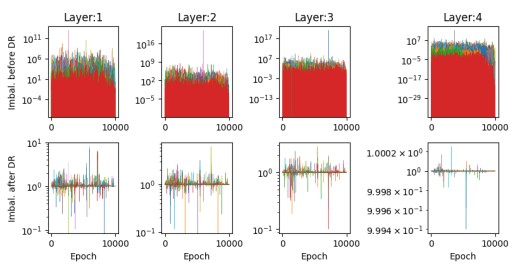

Figure 2: The degree of imbalance, i.e. the R.H.S. quantity of Eq. (5), before and after rebalancing every 10 epochs when training `roman-empire` on GAT. A value of 0 indicates complete balance.

Figure 3: A narrow range of larger learning rates results in better generalization for `roman-empire` dataset on GAT, with further improvement brought by dynamic rescaling (DR) to balance relative gradients.

This synergetic effect can be attributed to two factors. Firstly, several works report empirical evidence that larger learning rates are usually associated with flatter minima that have been linked to better generalization [49, 7, 19]. On the contrary, larger learning rates push the network out of its balanced state faster and more severely (see Fig. 2), which may impede trainability. However, this can now be addressed by rebalancing the network during training to facilitate trainability. Thus, balancing by dynamic rescaling supports higher learning rates in improving generalization. However, for a given task, only a narrow range of these 'large enough' learning rates can produce optimal results [24]. We show in Fig. 3 that training in balance can further improve the performance for this range of larger learning rates.

We use gradient clipping in combination with dynamic rescaling to accommodate exploding or vanishing gradients as a result of training with a larger learning rate and any numerical instabilities that may arise due to direct manipulation of parameter weights and gradients. We defer the ablation study of gradient clipping to the appendix.

### 3.2 Learning layers in order

As discussed in §2, one potential opportunity with dynamic rescaling to improve training dynamics is to control the order in which layers (or even neurons) in the network learn. We investigate this experimentally on both synthetic and real-world data, gaining novel insights into potential trends of optimal training dynamics for different tasks based on their homophily and heterophily. We defer the description of the synthetic data generation to the appendix.

The training curves and generalization performance achieved on the synthetic task for various train settings are summarized in Fig. 4. For this task, allowing the network to initially concentrate learning on the first layer results in faster training, lower minimum test loss, and better generalization than training the network in the standard setting or with dynamic rescaling. Interestingly, as more initial

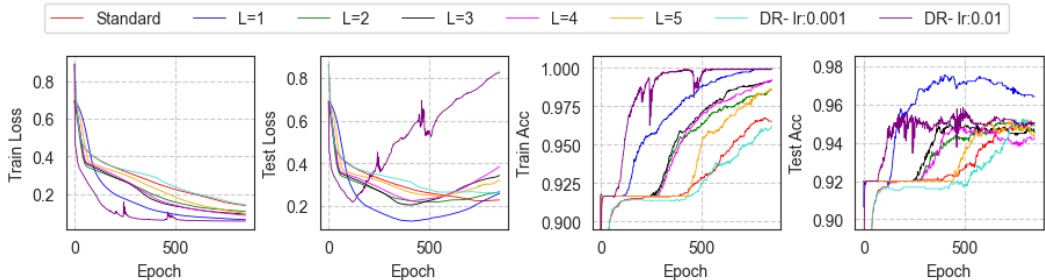

Figure 4: Performance of a five-layer GAT network on synthetic data under varying settings. Standard implies regular training with no constant or dynamic rescaling. $L = l$ for $l \in [5]$ denotes scaling down the parameters of $l$ by a constant ($\lambda = 0.002$) at initialization followed by regular training. DR denotes dynamic rescaling to balance relative gradients during training with the specified learning rate (lr). Note that the train and test accuracy axis have been zoomed in for clarity and the initial (train or test) accuracy is lower than 0.9 (but rises sharply in the first few epochs). The best strategy (among considered cases) for this task is to (initially) focus the learning more on the first layer.

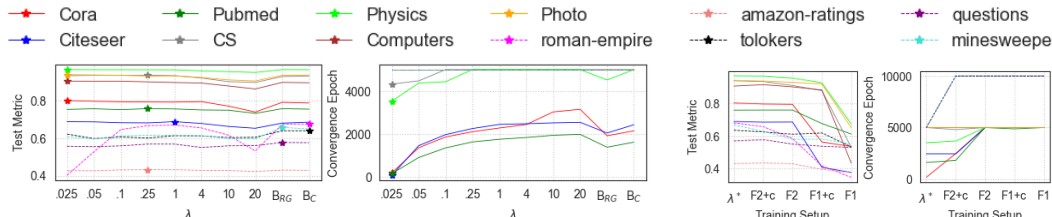

(a) The parameters $\mathbf{W}^1$ of the first layer are initially scaled by $\lambda$ in a loss invariant manner (i.e., $\mathbf{W}^2$ second layer is scaled by $\lambda^{-1}$). $\lambda = 1$ implies standard training. $B_{RG}$ and $B_C$ denote training in balance by $DR_{RG}$ and $DR_C$, respectively. Entries annotated with * indicate the best (highest) test metric and (lowest) convergence epoch. Models were trained for a maximum of 5000 epochs.

(b) $\lambda^*$ indicates the best case from (a). F1 and F2 denote the case where the first and second layers of the model are frozen, respectively, during training whereas +c indicates the presence of an additional linear classifier layer.

Figure 5: Impact of training layers of a two-layer GAT network in and out balance for different tasks. The tasks {`amazon-ratings`, `questions`, `roman-empire`, `tolokers`, `minesweeper`} are heterophilic and the remaining are homophilic. Homophilic tasks tend to perform better and converge much faster with learning concentrated in the first layer initially (lower weight norms imply larger relative gradients), whereas heterophilic tasks perform better when layers are trained in balance. Interestingly, even freezing the initial values of parameters in the second layer (i.e. only allowing the second layer to learn) does not significantly reduce the performance for homophilic tasks, even without an additional classifier layer. On the contrary, freezing the first layer results in a severe drop in performance for all tasks.

focus is placed on each subsequent layer, the training slows down and generalization worsens, with $l = 5$ being (marginally) the lowest. Dynamic rescaling with a learning rate of $0.001$, as used in all cases, is not as effective. However, using a larger learning rate of $0.01$ allows the fastest training with the second-best generalization. We expand on this as we analyze the evolution of relative gradients for interesting cases in Fig. 8 in the appendix.

Note that this synthetic task, which benefits most from concentrated learning in the first layer, is designed to be homophilic and thus differs from the heterophilic tasks that benefit more from training in balance. Prompted by this observation, we next study the impact of ordered learning of layers for real-world homophilic and heterophilic graphs.

As shown in Fig. 5, we find that real-world graphs indeed exhibit different trends regarding generalization and convergence time when trained in balance (w.r.t. relative gradients) and when trained out of balance (by controlling learning order of layers) depending on their homophilic or heterophilic nature. Training layers in balance is generally more effective for heterophilic tasks whereas allowing learning

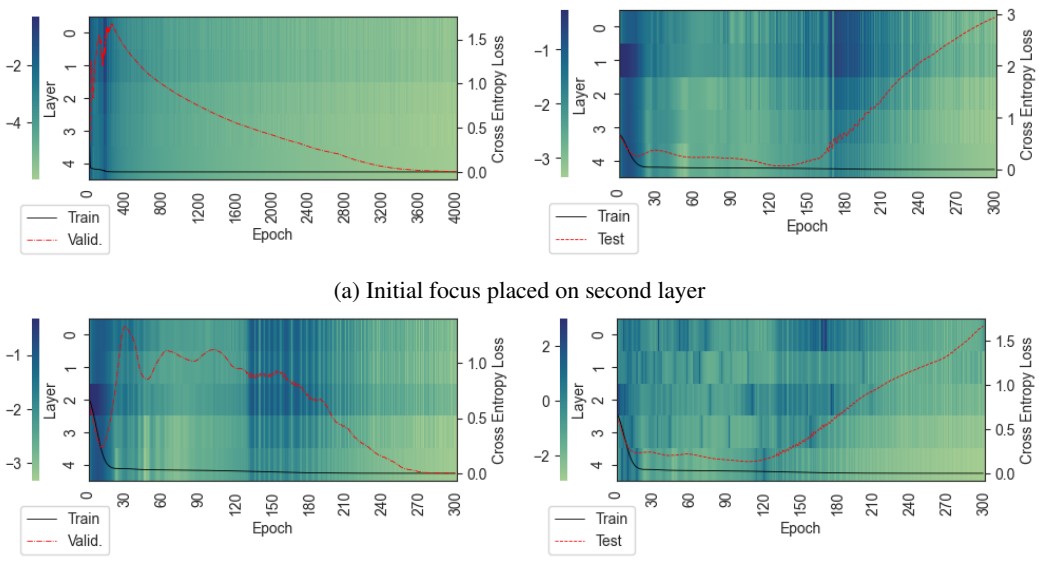

(a) Initial focus placed on second layer

(b) Initial focus placed on third (middle) layer

Figure 6: Evolution of gradient norms in a five-layer GCN network trained on synthetic data and evaluated on two validation sets from the same input distribution as the train set (left and right) with initial learning focused on the second and third layers in (a) and (b), respectively. The plot design is similar to that described in Fig. 8 except that in this case, the heatmap represents gradient norms.

to focus on the first layer tends to benefit homophilic tasks, particularly in terms of convergence time. We discuss this interesting observation more broadly in light of GNN training dynamics.

Generally, the optimal performance achieved by a model is, to a large extent, dependent on how well the inductive bias of the model architecture aligns with the task and its underlying graph structure. For example, it is widely known that general GNNs, without specially-introduced architectural elements, such as GCN perform better on homophilic than on heterophilic tasks. Intuitively, we hypothesize that homophilic tasks rely more on the neighborhood aggregation functionality of GNNs rather than feature learning. In this case, an aggregation over a random transformation of similar features may still be sufficient for good generalization.

Our insight is in line with a recent analysis of training dynamics [43] which shows that the NTK that controls the evolution of the learned GNN function tends to align with the message passing matrix (i.e. the adjacency matrix in most cases). Furthermore, for homophilic graphs, the adjacency matrix also aligns well with the optimal kernel matrix that represents nodes with the same label. As a result, on homophilic graphs, the alignment of the underlying structure with the optimal kernel matrix allows parameter-free methods similar to label propagation to perform at par with GNNs. However, the generalization of GNNs on heterophilic tasks, where the graph structure does not align with the optimal kernel, is mostly adversely impacted by the NTK aligning with the adjacency matrix. In other words, the structural information in the graph is not very relevant for a node's label in heterophilic settings and thus the node relies more on learning in the feature space rather than neighborhood aggregation. This is also supported by results showing that embedding additional MLP layers in the network significantly improves the performance of basic GNNs such as GATs on these heterophilic tasks [32]. Thus, we conclude that training in balance to potentially learn better feature transformations in all layers (and potentially neighbors farther away in deeper models) is more effective in heterophilic cases.

### 3.3 Grokking-like phenomena

Grokking [33] is defined as a phenomenon where the validation loss reduces, long after a near-zero train loss has been achieved, towards perfect generalization. It has been observed primarily in the context of algorithmic and synthetic datasets [30, 6, 41, 42, 22, 21, 11]. Yet, it is still regarded as a problem of fundamental interest, not only because the phenomenon appears to be puzzling at first,

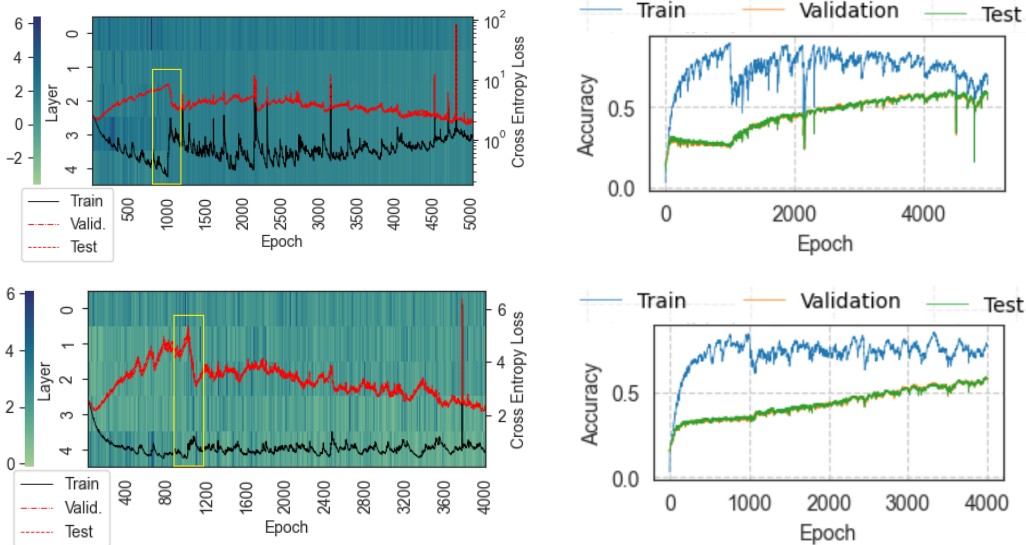

Figure 7: Left: Layer-wise relative gradient norm ($\log_{10}$ scale) and loss curves similar to Figure 3 in the paper. Right: Corresponding accuracy of the same run. Grokking-like phenomenon can be induced On a 5-layer GAT using real-world `roman-empire` dataset by placing initial training focus on layers 4 (top) and 5 (bottom) by scaling down initial parameter norms, followed by rebalancing w.r.t. relative gradients every 10 epochs staring only at epoch 1000. Note the sharp drop in validation/test loss immediately after rebalancing which also translates to more rapid improvement in test accuracy.

but also because it gives rise to nuanced insights into how neural networks learn, as it accentuates different training phases.

Grokking also touches upon our study in which order network layers learn best, as it has been attributed to network layers getting out of balance due to weight decay and weight norm decrease [22, 42]. More precisely, [14] argues that it is the result of delayed feature learning. According to their theory and observations for a 3-layer neural network, primarily the last layer learns first a linear combination of random features, as the parameter norms are scaled in such a way that it renders the first layer effectively untrainable, still reaching nearly zero training loss. With an increase of the last layer norm, however, feature learning by the first layer begins and starts to drive down the generalization error.

We conjecture that, depending on the learning task, grokking might be induced also by delayed learning of other layers, not only the last one. Our insights into rescaling based on relative gradients give us the means to test this hypothesis.

A synthetic node classification task is constructed, similarly as for the experiments in §3.2, except with $N = 100$, $p = 0.01$, and using a GCN instead of GAT as the target network. Using 50 nodes as the train set, we train a five-layer GCN and evaluate it on the remaining nodes as two disjoint test sets each of size 25.

Under the condition that the network is allowed to focus learning on the second or third hidden layer by scaling the weight parameters of the respective layer by $\lambda = 0.02$ at initialization, the resulting training curves in Fig. 6 exhibit a trend similar to grokking.

While one test set eventually achieves zero validation loss, the other's loss is reduced only to a certain extent before increasing again to give rise to a U-shaped curve that is more common. The latter is also similar to the only other case of grokking case detected in GNNs [23], to the best of our knowledge. Note that, in the former case, learning focused on the middle (third) layer of the network achieves perfect generalization much faster than when learning is focused on the second layer. This indicates that layer-level control of learning may be leveraged to facilitate faster generalization in grokking scenarios.

We also induce grokking-related behavior on a real-world dataset `roman-empire` (see Fig. 7) in two steps. Firstly, we allow only the last (or second to last) layer to learn which allows the training accuracy to increase continually while the test accuracy saturates or begins to drop. At this point, we rescale the network to bring all layers in balance w.r.t. relative gradients, following which, the test accuracy immediately begins to improve more rapidly accompanied by a drop in training accuracy. This can be interpreted as the network 'learning' more effectively when trained in balance rather than overfitting to the training data. While this is different from grokking where the training accuracy would generally not drop, it is independently an interesting observation on a real-world dataset.

Our key takeaway is that, as opposed to the general perception that grokking is induced by learning only the last layer, we observe a similar pattern by focusing learning on other (hidden) layers. Strikingly, grokking as a phenomenon can also support learning. While this possibility is a novel insight, we acknowledge that it is a potentially noisy phenomenon that materializes only in specific settings. We would like to highlight that understanding grokking is a separate area on its own and recent efforts are also limited to synthetic data [27]. Our induction of a similar phenomenon in real-world data by influencing learning dynamics can be of independent interest to develop a deeper theoretical understanding of such observations. Further investigation for more conclusive insights related to grokking is an interesting direction for future work.

## 4 Discussion

We have proposed to exploit the rescale invariance of GNNs such as GATs to control their training dynamics for improved training speed and generalization. To that end, we propose to rescale the parameters in such a way that the function, which is defined by a GNN that is trained, remains unperturbed, while the gradients are rescaled. The partial control of the learning dynamics, which is thus available, offers us the opportunity to teleport in the parameter space.

While the guiding principle of this teleportation is flexible, we propose to balance relative gradient norms such that all layers of the GNN are equally involved in the learning process. Our experiments highlight the potential of this rescaling to support larger learning rates and to balance relative gradients in search of flatter optima generally associated with better generalization. To our knowledge, we are the first to tap into the potential that the rescaling flexibility of GNNs has to offer.

While the goal of balancing relative gradients is to involve all network layers equally in learning, we have also analyzed the other end of the spectrum and discussed how we can influence the order in which GNN layers are learned. In doing so, we find that while training in balance is effective for heterophilic graphs, homophilic graphs tend to benefit more from training layers out of balance, with initially more focus on learning in the first layers. This novel insight into trends regarding training modes for different tasks contributes to understanding GNN learning dynamics under homophily and heterophily.

Furthermore, the ordered learning of layers also has implications for grokking-like phenomena, which provide insights into the inner mechanics of learning and are therefore also of fundamental interest. With our novel set of experiments, we could show that grokking does not necessarily result from a delay in feature learning, in which primarily the last layer learns in the first training phase. We could also induce a similar phenomenon by giving the middle layer a headstart in learning. Interestingly, this case provides also an example of a scenario in which a grokking-like phenomenon improves the overall generalization performance of the resulting model. We conjecture that unbalanced learning can also act as regularization that fights over-fitting.

Thus, our explorations and insights serve as the first stepping stone to developing a more comprehensive theory or set of guidelines to leverage dynamic rescaling for training not only GNNs, but various other deep learning architectures, to improve training speed, generalization, and, potentially, robustness.

## Acknowledgements

We gratefully acknowledge funding from the European Research Council (ERC) under the Horizon Europe Framework Programme (HORIZON) for proposal number 101116395 SPARSE-ML.

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

# A  Proofs

## A.1  Proof of Lemma 2.1

Lemma 2.1 states that, under the rescale invariance of GATs, if a parameter is scaled by $\tilde{\theta} = \lambda\theta$, then its gradient is scaled as $\nabla_{\tilde{\theta}}\mathcal{L} = \lambda^{-1}\nabla_{\theta}\mathcal{L}$.

*Proof.* We prove this by deriving the gradients of GAT network parameters.

Firstly, given input representations from layer $l-1$ and homogeneous activation functions $\phi_1$ and $\phi_2$, a GAT layer $l$ is defined as :

$$\mathbf{h}_v^l = \phi\left(\sum_{u \in \mathcal{N}(v)} \alpha_{uv}^l \cdot \mathbf{W}^l \mathbf{h}_u^{l-1}\right), \quad \text{where} \tag{8}$$

$$\alpha_{uv}^l = \frac{\exp(e_{uv}^l)}{\sum_{u' \in \mathcal{N}(v)} \exp(e_{uv}^l)}, \quad \text{and} \tag{9}$$

$$e_{uv}^l = (\mathbf{a}^l)^\top \cdot \phi_2(\mathbf{W}^l(\mathbf{h}_u^{l-1} + \mathbf{h}_v^{l-1})). \tag{10}$$

Given the following layer $l+1$

$$\mathbf{h}_v^{l+1}[k] = \phi_1\left(\sum_{u \in \mathcal{N}(v)} \alpha_{uv}^{l+1} \sum_i^{n_l} \mathbf{W}^{l+1}[k,i]\mathbf{h}_u^l[i]\right),$$

we derive the gradient of parameter $\mathbf{W}^{l+1}[k,j]$ w.r.t. to network loss $\mathcal{L}$ using the product rule:

$$\frac{\partial \mathcal{L}}{\partial \mathbf{W}^{l+1}[k,j]} = \frac{\partial \mathcal{L}}{\partial \mathbf{h}_v^{l+1}[k]} \frac{\partial \mathbf{h}_v^{l+1}[k]}{\partial \mathbf{W}^{l+1}[k,j]}$$

$$= \frac{\partial \mathcal{L}}{\partial \mathbf{h}_v^{l+1}[k]} \sum_{u \in N(v)}\left(\alpha_{uv}^{l+1}\mathbf{h}_u^l[j] + \sum_i^{n_l} \mathbf{W}^{l+1}[k,i]\mathbf{h}_u^l[i]\frac{\partial \alpha_{uv}^{l+1}}{\partial \mathbf{W}^{l+1}[k,j]}\right)$$

Now, if $\tilde{\mathbf{W}}^{l+1}[k,j] = \lambda^{-1}\mathbf{W}^{l+1}[k,j]$, then in order to keep $\mathcal{L}$ unchanged, $\tilde{\mathbf{W}}^l[j,i] = \lambda\mathbf{W}^l[j,i]$ and $\tilde{\mathbf{a}}^l[i] = \lambda^{-1}\mathbf{a}^l[i]$ which implies that $\tilde{\mathbf{h}}_u^l[j] = \lambda\mathbf{h}_u^l[j]$ (as derived in [28]) must hold so that $\tilde{\mathbf{h}}_v^{l+1}[k] = \mathbf{h}_v^{l+1}[k]$ remains unchanged, as:

$$\tilde{\mathbf{h}}_v^{l+1}[k] = \phi_1\left(\sum_{u \in \mathcal{N}(v)} \alpha_{uv}^{l+1} \sum_i^{n_l} \lambda^{-1}\mathbf{W}^{l+1}[k,i]\lambda\mathbf{h}_u^l[i]\right)$$

$$= \phi_1\left(\sum_{u \in \mathcal{N}(v)} \alpha_{uv}^{l+1} \sum_i^{n_l} \lambda^{-1}\mathbf{W}^{l+1}[k,i]\lambda\mathbf{h}_u^l[i]\right) = \mathbf{h}_v^{l+1}[k].$$

Consequently,

$$\frac{\partial \mathcal{L}}{\partial \mathbf{h}_v^{l+1}[k]} = \frac{\partial \mathcal{L}}{\partial \tilde{\mathbf{h}}_v^{l+1}[k]}.$$

We first show that:

$$\frac{\partial \alpha_{uv}^{l+1}}{\partial \tilde{\mathbf{W}}^{l+1}[k,j]} = \lambda\frac{\partial \alpha_{uv}^{l+1}}{\partial \mathbf{W}^{l+1}[k,j]}$$

For this, note that

$$e_{uv}^{l+1} = \mathbf{a}^{l+1}[k]\phi_2\left(\sum_i^{n_l} \tilde{\mathbf{W}}^{l+1}[k,i]\left(\tilde{\mathbf{h}}_v^l[i] + \tilde{\mathbf{h}}_u^l[i]\right)\right) + \sum_{m\neq i}^{n_l}\mathbf{a}^{l+1}[m]\phi_2\left(\sum_i^{n_l}\mathbf{W}^{l+1}[m,i]\left(\tilde{\mathbf{h}}_v^l[i]+\tilde{\mathbf{h}}_u^l[i]\right)\right)$$

$$= \mathbf{a}^{l+1}[k]\phi_2\left(\sum_i^{n_l}\lambda^{-1}\mathbf{W}^{l+1}[k,i]\left(\lambda\mathbf{h}_v^l[i]+\lambda\mathbf{h}_u^l[i]\right)\right) + \sum_{m\neq i}^{n_l}\mathbf{a}^{l+1}[m]\phi_2\left(\sum_i^{n_l}\mathbf{W}^{l+1}[m,i]\left(\lambda\mathbf{h}_v^l[i]+\lambda\mathbf{h}_u^l[i]\right)\right)$$

$$= \lambda\mathbf{a}^{l+1}[k]\phi_2\left(\sum_i^{n_l}\lambda^{-1}\mathbf{W}^{l+1}[k,i]\left(\mathbf{h}_v^l[i]+\mathbf{h}_u^l[i]\right)\right) + \lambda\sum_{m\neq i}^{n_l}\mathbf{a}^{l+1}[m]\phi_2\left(\sum_i^{n_l}\mathbf{W}^{l+1}[m,i]\left(\mathbf{h}_v^l[i]+\mathbf{h}_u^l[i]\right)\right).$$

and,

$$\frac{\partial\alpha_{uv}^{l+1}}{\partial\mathbf{W}^{l+1}[k,j]} = \frac{\partial\alpha_{uv}^{l+1}}{\partial e_{uv}^{l+1}}\frac{\partial e_{uv}^{l+1}}{\partial\mathbf{W}^{l+1}[k,j]}$$

$$= \frac{\partial\alpha_{uv}^{l+1}}{\partial e_{uv}^{l+1}}\left(\mathbf{a}^{l+1}[k]\left(\mathbf{h}_v^l[j]+\mathbf{h}_u^l[j]\right) + \sum_{m\neq i}^{n_l}\mathbf{a}^{l+1}[m]\phi_2\left(\sum_i^{n_l}\mathbf{W}^{l+1}[m,i]\left(\mathbf{h}_v^l[i]+\mathbf{h}_u^l[i]\right)\right)\right).$$

Then,

$$\frac{\partial\alpha_{uv}^{l+1}}{\partial\tilde{\mathbf{W}}^{l+1}[k,j]} = \frac{\partial\alpha_{uv}^{l+1}}{\partial e_{uv}^{l+1}}\frac{\partial e_{uv}^{l+1}}{\partial\tilde{\mathbf{W}}^{l+1}[k,j]}$$

$$= \frac{\partial\alpha_{uv}^{l+1}}{\partial e_{uv}^{l+1}}\lambda\left(\mathbf{a}^{l+1}[k]\left(\mathbf{h}_v^l[j]+\mathbf{h}_u^l[j]\right) + \sum_{m\neq i}^{n_l}\mathbf{a}^{l+1}[m]\phi_2\left(\sum_i^{n_l}\mathbf{W}^{l+1}[m,i]\left(\mathbf{h}_v^l[i]+\mathbf{h}_u^l[i]\right)\right)\right)$$

$$= \lambda\frac{\partial\alpha_{uv}^{l+1}}{\partial\mathbf{W}^{l+1}[k,i]}.$$

Also note that:

$$\frac{\partial\mathcal{L}}{\partial\tilde{\mathbf{h}}_v^{l+1}[k]} = \frac{\partial\mathcal{L}}{\partial\mathbf{h}_v^{l+1}[k]} \text{ since } \tilde{\mathbf{h}}_v^{l+1}[k] = \mathbf{h}_v^{l+1}[k].$$

Then, using all the above facts,

$$\frac{\partial\mathcal{L}}{\partial\tilde{\mathbf{W}}^{l+1}[k,j]} = \frac{\partial\mathcal{L}}{\partial\tilde{\mathbf{h}}_v^{l+1}[k]}\frac{\partial\tilde{\mathbf{h}}_v^{l+1}[k]}{\partial\tilde{\mathbf{W}}^{l+1}[k,j]}$$

$$= \frac{\partial\mathcal{L}}{\partial\mathbf{h}_v^{l+1}[k]}\sum_{u\in N(v)}\left(\alpha_{uv}^{l+1}\lambda\mathbf{h}_u^l[j] + \sum_i^{n_l}\mathbf{W}^{l+1}[k,i]\mathbf{h}_u^l[i]\frac{\partial\alpha_{uv}^{l+1}}{\partial\tilde{\mathbf{W}}^{l+1}[k,j]}\right)$$

$$= \lambda\left(\frac{\partial\mathcal{L}}{\partial\mathbf{h}_v^{l+1}[k]}\sum_{u\in N(v)}\left(\alpha_{uv}^{l+1}\mathbf{h}_u^l[j] + \sum_i^{n_l}\mathbf{W}^{l+1}[k,i]\mathbf{h}_u^l[i]\frac{\partial\alpha_{uv}^{l+1}}{\partial\mathbf{W}^{l+1}[k,j]}\right)\right)$$

$$= \lambda\frac{\partial\mathcal{L}}{\partial\mathbf{W}^{l+1}[k,j]}.$$

Therefore, scaling the parameter $\mathbf{W}^{l+1}[k,i]$ by $\lambda^{-1}$ scales its gradient by $\lambda = (\lambda^{-1})^{-1}$.

Next, we derive the gradients for parameter $\mathbf{a}^l[i]$.

$$\frac{\partial \mathcal{L}}{\partial \mathbf{a}^l[i]} = \frac{\partial \mathcal{L}}{\partial \mathbf{h}_u^l[i]} \frac{\partial \mathbf{h}_u^l[i]}{\partial \mathbf{a}^l[i]}$$

We note that under the rescaling, $\tilde{\alpha}_{uv}^l = \alpha_{uv}^l$. If $\tilde{\mathbf{a}}^l[i] = \lambda^{-1} \mathbf{a}^l[i]$, the rescale invariance implies that have also $\tilde{\mathbf{W}}^l[i,k] = \lambda \mathbf{W}^l[i,k]$ then:

$$
\begin{aligned}
\frac{\partial \mathcal{L}}{\partial \tilde{\mathbf{a}}^l[i]} &= \sum_{uv} \frac{\partial \mathcal{L}}{\partial \tilde{\mathbf{e}}_{uv}^l[i]} \frac{\partial \tilde{\mathbf{e}}_{uv}^l[i]}{\partial \tilde{\mathbf{a}}^l[i]} \\
&= \sum_{uv} \frac{\partial \mathcal{L}}{\partial \tilde{\mathbf{e}}_{uv}^l[i]} \phi_2(\tilde{\mathbf{W}}^l[i,:](\mathbf{h}_u^{l-1} + \mathbf{h}_v^{l-1})) \\
&= \lambda \sum_{uv} \frac{\partial \mathcal{L}}{\partial \tilde{\mathbf{e}}_{uv}^l[i]} \phi_2(\mathbf{W}^l[i,:](\mathbf{h}_u^{l-1} + \mathbf{h}_v^{l-1})) \\
&= \lambda \frac{\partial \mathcal{L}}{\partial \mathbf{a}}^l[i]
\end{aligned}
$$

Therefore, scaling the parameter $\mathbf{a}^l[i]$ by $\lambda^{-1}$ scales its gradient by $\lambda = (\lambda^{-1})^{-1}$.

This concludes the proof of Lemma 2.1.

$\square$

## B   Additional Details

**Synthetic task generation**   We construct a synthetic node classification task where we can examine the effects of allowing a layer to learn faster relative to others in a controlled environment. The input graph $G$ with $N = 5000$ is generated by the Erdős–Rényi (ER) model with edge probability $p = 0.001$. Input node features $\mathbf{h}_v^0 \in \mathbb{R}^d$ are sampled from a standard normal distribution. This input graph $G$ with no self-loops on nodes (i.e. $v \notin \mathbb{N}(v)$) is fed to a random Xavier [10] initialized GAT network $M_k$ of $k-1$ layers each of width $d$ and a final layer of width $C$, the number of classes of the classification task. This serves as the task's target network since its output embeddings are used to generate node labels in $G$. More precisely, for each node $v$, the node embedding output by $M_k$, $\mathbf{h}_v^{M_k}$ is effectively a function $f$ of nodes in the $k$-hop neighborhood $\mathbb{N}_k(v)$ of node $v$. $f$ is represented by a random GAT network as a series of non-linear transformations. Finally, we run $K$-means clustering on the neighborhood aggregated representation of nodes $\mathbf{h}_v^{M_k}$ to divide nodes into $C$ clusters. This clustering assigns node labels for our node classification task such that $y_v = \arg_{c \in [C]} (v \in c)$. We set $d = 10$, $C = 2$ and $k = 5$. This synthetic data generation ensures that the ground truth model can be represented by a GAT.

**Synthetic task training**   Given the input graph $G$ with a .75/.25/.25 train/validation/test split, we train a $L = k$ layer GAT network with the same architecture as $M_k$ but initialized with a looks-linear orthogonal structure which ensures that the network must learn the non-linear transformations of the target network. Furthermore, as noted in [28], the LL-orthogonal initialization facilitates trainability.

**Synthetic task results discussion**   We analyze the relative gradients of some interesting cases of training a five layer GAT on the synthetic task under various settings. As also observed in Fig. 4, the network generalizes best on the given task when the first layer (initially) learns faster than the other layers in Fig.8(c). This implies that the task demands more focus on the first layer, and thus enforcing all layers to learn (relatively) at the same rate in Fig.8(a) is not ideal. However, interestingly, a larger learning rate in Fig.8(b) enhances the network's ability to break free from its balanced state during training to have greater relative gradients in the first layer and decrease relative gradients in the latter layers with the last layer receiving the lowest. This also aligns with the pattern observed in Fig. 4. However, due to the dynamic rebalancing every 10 epochs, the network's favorable dynamic is still disrupted. This results in a smaller increase in test accuracy from Fig.8(a) as compared to the

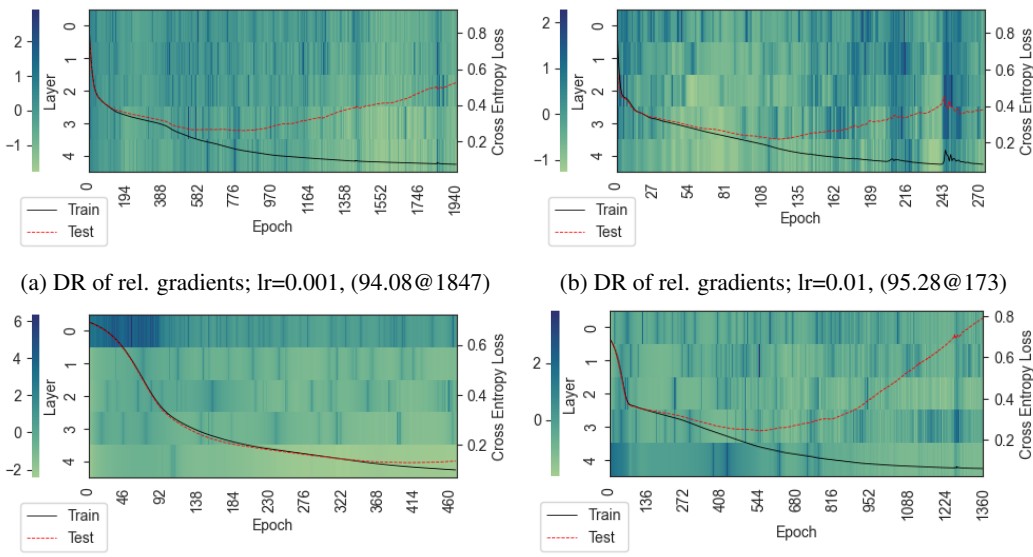

(a) DR of rel. gradients; lr=0.001, (94.08@1847)   (b) DR of rel. gradients; lr=0.01, (95.28@173)

(c) Init. scaling to focus on first layer; (97.04@367)   (d) Init. scaling to focus on last layer; (94.0@1260)

Figure 8: Evolution of relative gradients norms in a five-layer GAT network trained on synthetic data under varying settings. Test accuracy (%) @ epoch of maximum validation accuracy is reported in parentheses. The colored heatmap in the background displays $\log_{10}$ of relative gradient $l2-$norm for each layer (left axis) during training. Darker regions correspond to higher relative gradients. The training curves in the foreground show the train and test loss (right axis) for the epoch.

most favorable dynamics in Fig.8(c) for the task. Nevertheless, this supports our hypothesis that larger learning with dynamic rescaling potentially enables better generalization, as shown in Fig. 1 for real-world data. In comparison, initially restricting learning to only the last layer in Fig.8(d) is the worst case and comparable to enforcing all layers (including the last) to learn simultaneously. Although the test accuracy is high ($> 90\%$) in all cases, we focus on the relative difference in test accuracy under varying conditions, which are small, yet, significant as the goal of the task is to make the effects of different training conditions easily observable in isolation.

**Ablation of gradient clipping**   As evident from Table 2, employing gradient clipping alone does not improve the performance in the absence of dynamic rescaling.

Table 2: Results of training a 5-layer GAT network with and without gradient clipping (GC) in the case of no dynamic rescaling. The mean $\pm 95\%$ CI test metric at the epoch of the best validation metric across 10 splits is reported using the best learning rate from $\{0.01, 0.001, 0.005\}$. The evaluation metric is accuracy for `roman-empire` and `amazon-ratings`, and ROC AUC for the remaining datasets.

|              | roman-empire | amazon-ratings | tolokers | questions | minesweeper |
|--------------|--------------|----------------|----------|-----------|-------------|
| w/o DR w/o GC | 0.4978 ± 0.0209 | 0.4545 ± 0.0043 | 0.6493 ± 0.008 | 0.5829 ± 0.0172 | 0.5057 ± 0.0058 |
| w/o DR w/ GC  | 0.3711 ± 0.0357 | 0.4535 ± 0.0025 | 0.6441 ± 0.0116 | 0.5745 ± 0.0135 | 0.5019 pm 0.0092 |

**Limitations**   We elaborate on the current limitations of this work as follows:

- The primary limitation of dynamically rescaling a model is that we require the rescale invariance of the model architecture (if it exhibits one), which may vary widely across different GNN architectures. The dynamic rescaling proposed in the paper applies to GCNs and GATs, two standard baseline GNN models commonly used to study and develop

insights into the training dynamics of GNNs. While in theory, this is a prerequisite to dynamically rescale the network during training without altering its function to potentially improve training dynamics, we conduct an experiment where this prerequisite is not fulfilled by simply replacing the ReLU activation with the Tanh activation that is not positively homogeneous. Consequently, the network no longer exhibits rescale invariance. However, we find that in practice, the advantages of dynamic rescaling to train in balance can still be observed in terms of better generalization. For example, on the roman-empire dataset, using Tanh activation in a 5 layer GAT trained on roman-empire achieves an average test accuracy (over 3 runs) of $58.14 \pm 4.64\%$ and $30.98 \pm 2.32\%$ with and without rebalancing, respectively. While this trend aligns with that observed using ReLU, the training may be more noisy as each rescaling during training is not loss invariant.

- From an implementation perspective, directly manipulating model parameter values and gradients during rescaling can result in numerical instability issues that we currently regulate using gradient clipping, a commonly used practice in machine learning. Nevertheless, a more principled approach to tackle this problem could be beneficial.

- Rebalancing repeatedly and frequently during training may incur computational overhead in practice. However, it may be offset by the increased training speed, requiring fewer epochs overall. From a time complexity perspective, rebalancing only adds a constant factor of operations in each epoch determined by the number of iterations in one rebalancing step. In practice, we find that only a few iterations ($< 10$) are necessary to balance the network approximately enough to gain better generalization and/or training speed.

