# OpenReview forum: "Dynamic Rescaling for Training GNNs"
_NeurIPS.cc/2024/Conference — NeurIPS 2024 poster_

### Official Review · Reviewer_GhMV · 2024-06-30

**Soundness:** 3
**Presentation:** 3
**Contribution:** 3
**Rating:** 6
**Confidence:** 4

**Summary:**

This paper proposes the dynamics rescaling approach to improve the trainability of GAT. This paper is motivated by the rescale invariance property of GAT, i.e. the function is invariant w.r.t. some scaling terms on a neuron’s weights, and the corresponding conservation law. To balance the network throughout training, this paper proposes a criterion (inspired by the conservation law) to dynamically rescale weights. This approach enables the network to benefit from larger learning rates while circumventing the imbalance problem, and thus improves generalization. The paper also discusses other use cases of the approach, such as enabling layer-wise training to accelerate grokking.

**Strengths:**

This paper studies training dynamics of GNNs, which is an important yet underexplored topic in graph learning. The proposed method is well-motivated from a symmetry perspective, and can effectively improve the trainability and generalizability of GAT by forcing balance. The empirical observations from controlling the order in which layers are trained are also intriguing and might be of independent interest in other areas.

**Weaknesses:**

1. As preliminary is integrated into the approach, it is hard to parse which part is novel. It would be beneficial to more clearly compare the contributions of this paper and previous papers on training dynamics, especially [1].
2. The experiment in sec.4.1 is relatively weak as the considered five datasets have similar properties (heterophily). It is also questionable if the proposed method can work on GNNs that do not satisfy the invariance property.
3. It is not entirely clear how the proposed criterion is related with the conservation law associated with GAT.

[1] Are GATS out of balance? in NeurIPS 2023

**Questions:**

Are there any experiments on steering the model to learn certain layers first on real-world datasets to test if this idea of accelerating grokking is actually useful in practice? Particularly, I wonder if developing methods to induce feature learning (or grokking) is really effective for improving the peak performance beyond synthetic datasets, as there are also results suggesting proper way of learning in the kernel regime (or lazy learning regime) is sufficient for good generalization [2]. It would be interesting to see discussions in this direction.

[2] How Graph Neural Networks Learn: Lessons from Training Dynamics. in ICML 2024

**Limitations:**

While the analytical framework might be generally useful, but the method seems only applicable to those GNNs that satisfy the rescale invariance property.

---

> ### Author Rebuttal · Authors · 2024-08-07
>
> We thank the reviewer for acknowledging the importance of our work and their constructive feedback. We address their concern as follows. Note that all references to figures are in the PDF attached to the global response unless stated otherwise.
>
> 1. We gladly highlight the novelty of our work in comparison to past work. To the best of our knowledge, we are the first to propose dynamic rescaling as a tool to influence the training dynamics of GNNs that can be used in various ways. For example, ensuring all layers have an equal opportunity to drive change in their parameters by training in balance or allowing a particular layer with higher relative gradients to train `more' than others. Particularly concerning [1], our work differs in two ways:
>
> * While [1] is the first work to evaluate the role of network balance in the generalization and training speed of GATs, their method is restricted to balance at initialization. In theory, this balance would be maintained throughout training under gradient flow (i.e. gradient descent with infinitesimal learning rates). However, in practice, as a result of training with finite learning rates (potentially in a range of larger values associated with better generalization - see Fig. 1), the network balance established by the proposed initialization of [1] is disrupted. We show the degree of imbalance in the network during training before and after the rebalancing of the network every 10 epochs in Fig 2. The balanced initialization of [1] thus cannot ensure that the network will stay in balance throughout training. For this, we provide a method using dynamic rescaling and demonstrate the benefits of training GNNs in balance, specifically for heterophilic tasks.
>
> * We challenge the balancing criterion based on norms of parameter values (used in [1]) and instead propose balancing w.r.t. the norm of relative gradients of parameters. Table 1 in the paper shows that relative gradients are often the better criterion for improved training resulting in better generalization.
>
> 2. The reviewer has rightly pointed out that a primary limitation of our approach is that we need to know the rescale invariance of the considered GNN architecture. In theory, this is a prerequisite to dynamically rescale the network during training without altering its function to potentially improve training dynamics. However, we conduct an experiment where this prerequisite is not fulfilled by simply replacing the ReLU activation with the Tanh activation that is not positively homogeneous. Consequently, the network no longer exhibits rescale invariance. However, we find that in practice, the advantages of dynamic rescaling to train in balance can still be observed in terms of better generalization. For example, on the roman-empire dataset, using Tanh activation in a 5 layer GAT trained on roman-empire achieves an average test accuracy (over 3 runs) of $58.14\pm4.64\%$ and $30.98\pm2.32\%$  with and without rebalancing, respectively. While this trend aligns with that observed using ReLU, the training may be more noisy as each rescaling during training is not loss invariant.
>
> 3. The conservation law of GAT describes a relationship between the norms of parameters associated with each neuron that the network adheres to under gradient flow. This law is induced by the rescale invariance of the network parameters. Thus, we can rescale the network parameters to fulfill any criterion involving parameter values or norms. One such criterion is the specific sum of parameter l2-norms defined by the conservation law during training. However, as we prove in Lemma 1 in the paper, the parameter gradients also depend on the scaling factor of the parameter values, and thus can also be controlled directly or indirectly by the scaling factors. This allows us to opt for a criterion that depends both on the parameter values and gradients (such as relative gradients) that can be fulfilled by rescaling without changing the network function. However, unlike the criterion based on parameter L2-norms, the proposed criterion of relative gradients is not naturally fulfilled during training and thus requires us to rebalance the network frequently.
>
> 4. As the reviewer has rightly noted, the main advantage of training GATs in balance is observed for heterophilic tasks, while for homophilic tasks such as on the Cora and Citeseer datasets, there is no significant impact of training in balance w.r.t. relative gradients which perform similarly to standard training. Interestingly, we can relate this observation to the reviewer's question regarding results in [2], which suggests that a proper way of learning in the kernel regime (or lazy learning regime) is sufficient for good generalization in the case of homophilic datasets.  A detailed discussion of this insight with empirical verification, as well as an example of inducing a grokking-like phenomenon on a real-world dataset, is presented in the global response. Thus, we request the reviewer to read the global response and consider it a part of our response to them.

---

> > ### Comment · Reviewer_GhMV · 2024-08-12
> >
> > Thank you for the response. My concerns are addressed. I believe this paper has made valuable contributions to the understanding of GNNs' optimization. I keep my score.

---

### Official Review · Reviewer_uEaK · 2024-06-30

**Soundness:** 2
**Presentation:** 2
**Contribution:** 2
**Rating:** 4
**Confidence:** 4

**Summary:**

This manuscript attempts to study the rescale invariance of GNNs and suggests to dynamically rescale the network using relative gradient, i.e. g/theta, where g is gradient of theta. Several experiments are conducted to to show how the dynamic rescaling using relative gradient norms affects the learning speed and generalization.

**Strengths:**

- this paper is the first propose to use relative gradient norm to study the invariance of GNNs.
- several interesting experimental phehomenon have been identified, such as the training speed accross the different layers for the grokking

**Weaknesses:**

- From algorithimic side, my main concern is that the proposed dynamical scaling does not induce a consistent improvement over the tasks the authors have considered. This brings the skeptism whether this strategy works or not.
- If we did not treat this paper as algorithm paper, instead, both the theoretical and empirical analysis, as well as potential insights, should be in more depth.
So, I think this paper initializes a good idea and point to start with, but less developed. It is still not mature yet to be published.

**Questions:**

See the above

**Limitations:**

See the above

---

> ### Author Rebuttal · Authors · 2024-08-07
>
> We thank the reviewer for acknowledging the novelty of our work and their interest in our insights. We address their concerns as follows.
>
> We would like to reiterate that our goal is not to propose a one-size-fits-all solution or achieve state-of-the-art performance. Rather, this work aims to conduct an exploratory study of a novel idea and gain insights into GNN learning dynamics and Grokking-like phenomena. In doing so, we highlight our main contributions as follows.
>
> 1. We prove that if GNNs exhibit a rescale invariance (like GCNs or GATs), we can manipulate gradient norms while leaving the GNN and the loss invariant and thus influence the learning dynamics.
> 2. Utilizing this insight, we propose to dynamically rescale GNNs based on a novel criterion, i.e., relative gradient norms, to foster trainability and enable potentially faster training with larger learning rates.
> 3. We derive a synthetic learning task that supports our insights into learning dynamics and provides novel insights into Grokking.
> 4. Our methodological contributions lead to a couple of novel insights, which are discussed next.
>
> As requested by the reviewer, we further strengthen our insights and deepen our empirical analysis with the support of additional experiments, as follows.
>
> 1. We discover trends in optimal training dynamics regarding homophily and heterophily, which aligns with a recent analysis of learning dynamics in general GNNs. In summary, training GNNs in and out of balance may be more beneficial for heterophilic and homophilic tasks, respectively.
>
> 2. We construct a grokking-like phenomenon on real-world data, which we can induce by a combination of different ways to influence gradient dynamics. Interestingly, a worsening training accuracy can be associated with an improving test accuracy, suggesting that training GNNs in balance can potentially promote better feature learning where required (such as in heterophilic tasks) and mitigate over-fitting.
>
> 3. While we show in the main paper that dynamic rescaling allows better generalization with larger learning rates, we extend the experiments to cover a wider range of learning rates and observe the trend to be in line with [1].
>
> Due to space limitations and overlap with questions from other reviewers, we have presented a detailed discussion of these insights in the global response. We request the reviewer to please also consider the global response a part of our response to them.
>
> [1] Lobacheva et al. Large Learning Rates Improve Generalization: But How Large Are We Talking About? (2023).

---

> > ### Author Response · Authors · 2024-08-14
> > **Request for Response**
> >
> > We kindly request the reviewer to respond to our rebuttal soon due to the limited duration of the remaining discussion period. We would be happy to answer any further questions the reviewer may have.

---

### Official Review · Reviewer_TkPU · 2024-07-10

**Soundness:** 3
**Presentation:** 1
**Contribution:** 2
**Rating:** 3
**Confidence:** 3

**Summary:**

This paper investigates the use of dynamic rescaling to train Graph Attention Networks (GATs), a type of Graph Neural Network (GNN). The method aims to enhance the trainability and generalization of GATs by balancing network parameters and gradients during training. GATs' rescale invariance property enables them to adjust the rate at which network layers learn, resulting in better training dynamics optimization. The proposed dynamic rescaling technique balances network parameters using their relative gradients, resulting in faster and more efficient training. Higher learning rates can be achieved by dynamically rebalancing the network during training, resulting in better generalization on real-world data.
The document emphasizes the importance of selecting suitable rescaling criteria and procedures based on network parameters and gradients. It emphasizes the potential benefits of dynamic rescaling in controlling the order of learning among network layers, which can affect the overall performance of GATs. Experimental studies are being conducted to validate dynamic rescaling's effectiveness in improving GAT training speed, generalization, and robustness.
Furthermore, the paper discusses related research on training dynamics and generalization in GNNs, demonstrating how dynamic rescaling can outperform traditional methods such as initialization, normalization, and regularization. The findings indicate that leveraging the rescale invariance of GNNs, such as GATs, can result in sharper optima, better generalization, and greater robustness during training. The paper concludes by discussing grokking-like phenomena associated with learning patterns in GNNs and highlighting promising future research directions in the use of dynamic rescaling for more practical benefits in GNN optimization.

**Strengths:**

The paper focuses on a compelling topic and observes several notable phenomena, including: 1. A balanced state during training combined with larger learning rates results in better generalization compared to when either component is used alone. 2. The Grokking-like phenomena.

**Weaknesses:**

In general, I appreciate the topic of this work, which focuses on the training dynamics of graph neural networks and aims to improve them. However, the following points indicate that it is not yet ready for publication:

1. Presentation: I have included questions 1 to 4 in the next section. While I am aware of the answers to these questions, I suggest the authors use them as references to clarify the corresponding parts of the draft, thereby improving its readability.

2. Analysis: More interesting results might be obtained by examining the connection between learning, grokking, and the balanced state. I recommend that the authors conduct an in-depth analysis of this connection.

**Questions:**

1. Line 5: Does a larger learning rate help generalization? If larger learning rates tend to induce improved generalization but disrupt the "balanced state," what is the effect of the balanced state? Does this mean it is not beneficial for generalization?

2. Line 22: To improve the readability of this work, I suggest the authors assume readers generally do not have extensive knowledge about the "conservation law" and "conservation of a relationship between network parameters and gradients." Could you please use one more sentence in paragraph 2 to clarify these concepts?

3. Line 31: Please correct the citation format.

4. Why do we need balance? Is it for the trainability of GNNs? If so, I suggest you accurately describe what trainability means. Otherwise, in the next paragraph, starting with “The common criterion used to balance networks,” it seems like you already assume readers are fully convinced of the need for balance.

**Limitations:**

I would like to encourage the authors to make a separate Limitations section in their paper.

---

> ### Author Rebuttal · Authors · 2024-08-07
>
> We thank the reviewer for the constructive feedback and address their concerns as follows. Note that all references to figures are in the PDF attached to the global response unless stated otherwise.
>
> 1.  Firstly, to improve the readability and comprehension of the paper, we answer the questions set forth by the reviewer and provide a holistic background and discussion of conservation laws, trainability, and large learning rates. We will include this in the updated version of the paper.
>
> * **Conservation law and balance:** It is known that, for traditional deep feed-forward networks (DNNs) and convolutional neural networks (CNNs) with homogenous activation functions such as ReLUs, the difference between the squared l2-norms of incoming and outgoing parameters to a neuron stays constant (and is thus conserved) under gradient flow, i.e. gradient descent with an infinitesimal learning rate. Such conservation laws arise for neural networks exhibiting a rescale invariance property [7]. When this conserved quantity is (nearly) zero, the network is said to be in a balanced state. The concept of balance at the neuron level is rooted in an understanding of the training dynamics of deep neural networks where it is generally assumed that norm balance induces conditions for successful training. We have discussed more implications of the conservation law for DNNs and CNNs in the related work. In the context of GNNs, the insight regarding norm balance was presented by [6], which derives the conservation law for GATs and demonstrates how a network initialization fulfilling this balance property, in practice, enhances the trainability that we discuss next.
>
> * **Trainability:** Trainability can be defined as the ability of model parameters to change during training from their initial state to an optimal position. In general, the (relative) gradient norm of parameters serves as a good proxy for this measure, as it quantifies the relative update of parameters in each training epoch, and has been used in literature to study trainability [6,8]. As mentioned above, a balanced initial state enhances trainability [6]. In theory, under gradient flow due to the conservation law, this balanced state should hold throughout training. However, in practice, using finite (and larger) learning rates disturbs this balance. We propose a novel idea of dynamic rescaling of the network to rebalance the network during training. Additionally, we show that norm balance may not be the ideal criterion for better trainability and propose balancing the network w.r.t. relative gradients of parameters instead. Intuitively, this allows all parameters an equal opportunity to change during training and thus improves trainability.
>
> * **Larger learning rates:** As we state in the paper, several works report empirical evidence that a larger learning rate biases toward flatter minima which have been linked with better generalization [2-5]. The same idea is endorsed by [1], which also states that only a narrow range of these `large enough' learning rates can produce optimal results. However, as discussed above, larger learning rates cause the network state to become more imbalanced during training and thus may impede learning. We show that rebalancing the network during training by dynamic rescaling further improves the generalization brought about by larger learning rates (see Fig. 1).
>
> 2. Secondly, as suggested by the reviewer, we elaborate on our insights relating learning, balanced state, and grokking-like phenomena in further depth with the support of additional experiments. Due to space limitations and overlap with questions from other reviewers, we present these further insights in the global response. We request the reviewer to please also consider the global response a part of our response to them.
>
> 3. Finally, we include an elaborate discussion of the limitations of this work also in the global response.
>
> [1] Lobacheva et al. Large Learning Rates Improve Generalization: But How Large Are We Talking About? (2023).
>
> [2] Zhao et al. Penalizing gradient norm for efficiently improving generalization in deep learning (2022).
>
> [3] Lewkowycz et al. The large learning rate phase of deep learning: the catapult mechanism. (2020).
>
> [4] Seong et al. Towards Flatter Loss Surface via Nonmonotonic Learning Rate Scheduling. (2018).
> [5] Dinh et al. Sharp minima can generalize for deep nets. (2017).
>
> [6] Mustafa et al. Are GATs Out of Balance? (2023).
>
> [7] Kunin et al. Neural Mechanics: Symmetry and Broken Conservation Laws in Deep Learning Dynamics. (2021)
>
> [8] Jaiswal et al. Old can be Gold: Better Gradient Flow can Make Vanilla-GCNs Great Again. (2022).

---

> > ### Comment · Reviewer_TkPU · 2024-08-11
> >
> > Thank you for addressing my questions point by point. I have reviewed your global response, and while I appreciate your efforts, I believe the current version of the submission still requires significant improvements before it can be accepted. The topics of learning rate and grokking, in particular, deserve a more thorough discussion rather than being mentioned only in passing as interesting points. Besides, the motivation behind your work should be clarified more effectively, as the current explanation in the fourth paragraph does not sufficiently convey its significance.
> > I encourage the authors to consider the feedback provided as suggestions to enhance the readability and depth of the paper. I look forward to seeing your work in a future conference.

---

> > > ### Author Response · Authors · 2024-08-11
> > >
> > > Thank you for your response. As the reviewer has acknowledged our point-by-point answers to the questions that were put forth as a guideline for us to enhance the readability, and will eventually be included in the revised version of the paper, we believe this concern has been addressed comprehensively.
> > >
> > > Regarding the depth of this work and the topics of learning rate and grokking, we wish to clarify our motivation and contributions, as follows, in light of which we would be happy to revise the introduction section (particularly paragraph 4 as suggested by the reviewer) to make the motivation more clear.
> > >
> > > 1. Primarily, we prove that if GNNs exhibit a rescale invariance (like GCNs or GATs), we can manipulate gradient norms while leaving the GNN and the loss invariant and thus influence the learning dynamics. This is motivated by earlier studies on deep feedforward non-linear neural networks that exploit rescale invariance, using transformations respecting loss-invariant symmetry to teleport parameters to another point in the loss landscape with steeper gradients to improve optimization and/or convergence [10]. To the best of our knowledge, we are the first to explore this concept in GNNs, where identifying and exploiting the rescale invariance is not as straightforward, due to the different architectural elements such as node-wise neighborhood aggregation. This is also discussed in lines 163-171 in the paper.
> > >
> > > 2. Secondly, motivated by the positive outcomes of training a model balanced at initialization as shown by [6], we utilize our insight to explore the effects of maintaining this balance throughout training rather than only at initialization, as enabled by our derivation of a procedure to balance a GAT network w.r.t. a criterion that is a function of network parameters and gradients by dynamic rescaling.
> > > 3. In addition, we challenge the balancing criterion based on norms of parameter values (used in [6]) and instead propose balancing w.r.t. the norm of relative gradients of parameters. Table 1 in the paper shows that relative gradients are often the better criterion for improved training resulting in better generalization.
> > >
> > > 4. We realize another possible way to use the rescale invariance (and consequent dynamic rescaling) is to control the relative order in which layers learn arbitrarily, at any point in time during training.
> > >
> > > As the reviewer has pointed out, the topics of learning rate and grokking, and their relationship to learning dynamics, indeed merit further study. However, it is currently out of the scope of this work as these topics are not our primary subjects. Our work is instead focused on the methodological contributions outlined above that lead to various novel insights. Specifically regarding graph learning, we discover potential trends of optimal learning dynamics for homophily and heterophily, and more generally regarding learning rates and interesting grokking-like phenomena that we observe empirically and thus report in our discussions. These insights can be built upon to further explore the relationship between grokking and learning rates by ourselves and the community. We would like to highlight that understanding grokking is a separate area on its own and recent efforts are also limited to synthetic data [9]. Our induction of a similar phenomenon in real-world data by influencing learning dynamics can be of independent interest to develop a deeper theoretical understanding of such observations.
> > >
> > > We sincerely appreciate the interest and positive outlook of the reviewer and would highly welcome suggestions on how we could improve this work within its scope. We thank them greatly for their valuable time and efforts.
> > >
> > > [9] Mohamadi et al. Why Do You Grok? A Theoretical Analysis on Grokking Modular Addition. (ICML 2024).
> > >
> > > [10] Zhao et al. Symmetry Teleportation for Accelerated Optimization. (NeurIPS 2022).

---

### Official Review · Reviewer_2KpP · 2024-07-11

**Soundness:** 3
**Presentation:** 3
**Contribution:** 3
**Rating:** 7
**Confidence:** 3

**Summary:**

The paper studies the various phenomena in GNN training that prevent the usage of large learning rates for faster convergence and better generalization. Based on the theoretical foundation that large learning rates can be stably used only when the rescale symmetry of the loss function is satisfied, the authors propose a way to systematically perform a dynamic rescaling on the network parameters with relative gradient norms to maintain the rescale symmetry throughout training. This allows the usage of large learning rates leading to better convergence and generalization. Experiments reveal that their proposed approach for dynamic rescaling leads to consistently better empirical generalization across multiple benchmarks, as well as interesting phenomena akin to grokking arising from the arbitrary controllability of the order in which layers are learnt as a consequence of their proposed rescaling mechanism.

**Strengths:**

1. The paper presents a novel insight into the problem of convergence of GNNs trained with large learning rates. Although large learning rates are known to provide better generalization, convergence while using large learning rates still remains a challenge, hindering its adoption in practice. The authors interestingly observe that certain factors that get introduced while training neural networks in practice, such as momentum, weight decay, batch stochasticity break the rescale symmetry of the loss function. In fact, better generalization from large learning rates can only be achieved under rescale symmetry. However, the breaking of the rescale symmetry is the core factor hindering the usage of large learning rates while still achieving stable training dynamics.

2. To remedy the above problem, the authors introduce a dynamic rescaling mechanism, where they periodically rescale the network parameters with relative gradient norms to explicitly maintain rescale symmetry and consequently, the corresponding conservation law throughout training, thereby allowing the usage of large learning rates, while maintaining trainability and achieving generalization.

3. The dynamics rescaling mechanism allow the authors to arbitrarily determine the order in which network layers are trained, which provides greater flexibility while designing learning schemes, especially for OOD generalization tasks.

4. The experiments show that dynamic rescaling of network parameters with the proposed relative gradient norms achieves consistently improved generalization on multiple benchmarks.

5. The authors present interesting experiments with their methodology that illustrate grokking like phenomenon while explicitly controlling layer learning order. Interestingly, they observe that grokking might also be induced by the learning of other intermediate layers, and not just the last.

**Weaknesses:**

1. Line 8, Abstract is incomplete: "on relative gradients, that promotes faster and better." I suppose the authors meant convergence.

2. What is exactly meant by "larger learning rates tend to induce improved generalization but make the training dynamics less robust"? More specifically, what do the authors mean by "robustness" in the context of training dynamics? Throughout the paper, the authors talk about improvement in training speed and generalization, but none of them necessarily imply "robustness of the training dynamics".

3. Enforcing Lipschitz continuity (to prevent vanishing / exploding gradients) through gradient clipping introduces a sharp penalty leading to instability in the optimization process, whereas gradient penalty is known to produce smoother training dynamics, at least in the context of training WGAN critics [a]. This result is not restricted to WGANs, since gradient clipping, by definition introduces a sharp cut-off, while gradient penalty is inherently a smoother regularization constraint. Then, why do the authors go for gradient clipping? A discussion is necessary.

4. The paper can benefit from a more elaborate discussion of the limitations, which are not very clear from the current version.

5. Minor: Line 110: Comma at the beginning of the sentence.

References:

[a] Gulrajani et al, "Improved Training of Wasserstein GANs", NeurIPS 2017.

**Questions:**

Please refer to the Weaknesses section.

**Limitations:**

The paper can benefit from a more elaborate discussion of the limitations, which are not very clear from the current version.

---

> ### Author Rebuttal · Authors · 2024-08-07
>
> We thank the reviewer for the constructive feedback and address their concerns and questions as follows.
>
> 1. Thanks for pointing out this typo. We meant training. We would like to further clarify the statement in the abstract regarding larger learning rates and robustness. Here, by robustness, we mean general training stability. Larger learning rates in neural networks are generally associated with better generalization but more unstable training [1]. We state this already observed behavior from literature. Our finding is that dynamic rebalancing of relative gradients further improves the generalization obtained by using larger learning rates. For reference, please see Figure 1 in the attached PDF in the global response. As the reviewer has rightly pointed out, we do not discuss the robustness (to noise, or adversaries) aspect of dynamic rescaling in this work, and using the term training stability would be a better alternative.
>
> 2. We opted for gradient clipping, as it is a widely used standard practice, generally used across various domains of machine learning to avoid exploding gradients. However, upon the reviewer's valuable suggestion, we have also tried to replace gradient clipping with an L2-norm gradient penalty. As evident from Table 1 below, we find that the performance is adversely impacted, and gradient clipping (even in combination with gradient regularization) performs better. Nevertheless, it merits a discussion in the related work that we will include.
>
> Table 1: Mean test accuracy (\%) over 10 runs for \texttt{roman-empire} dataset on GAT using gradient l2-norm penalty (of strength $\lambda$) with and without gradient clipping (GC) while rebalancing w.r.t relative gradients every epoch during training. Rebalancing only in combination with gradient clipping without using gradient penalty results in $59.62\pm2.21\%$ mean test accuracy.
> | $\lambda$ |           0.01              |          0.1               |            2                 |           10                 |
> | :----------: | :--------------------: | :--------------------: | :-------------------: | :--------------------: |
> | w/o GC       | $50.43 \pm 3.07$ | $52.76 \pm 2.75$ | $51.4 \pm 3.42$ | $52.44 \pm 2.43$ |
> | with GC      | $\mathbf{59.28 \pm 2.55}$ | $\mathbf{59.82 \pm 2.64}$ | $\mathbf{58.4 \pm 1.85}$ | $\mathbf{59.7 \pm 2.35}$ |
>
> 3. We include an elaborate discussion of the limitations of this work in the global response.
>
> [1] Lobacheva et. al. Large Learning Rates Improve Generalization: But How Large Are We Talking About? (2023)

---

> ### Comment · Reviewer_2KpP · 2024-08-12
>
> I thank the authors for their response and the set of experiments that they have conducted. Although some of my concerns have been clarified, I will need some additional deliberation on my end to decide upon my final score.
>
> Meanwhile, I still have one major concern about the experiment on the prevention of exploding gradients presented in Table 1 of the rebuttal. In the (second) "with GC" row, if I understand correctly, the authors apply both gradient penalty and gradient clipping. Although with only gradient penalty, i.e., the first row, there is some drop in performance from the results reported in Table 1 of the main paper, of around 2% from 54% to 52%), applying both gradient penalty and clipping in conjunction leads to a significant boost of around 5% (from 54% to 59%) relative to the authors' original results. Is my understanding correct? If so, then it may be unfair to state that "the performance is adversely impacted" with the use of gradient penalty, since in conjunction with GC, it gives a significant boost.
>
> Since the approach involves dynamic rescaling wrt relative gradient norms, I believe developing an understanding of the interactions between the rescale operations and the regularization is key for the soundness of the central claims, something that is lacking from the work and the rebuttal in their current state.

---

> > ### Author Response · Authors · 2024-08-13
> >
> > Thank you for your response. Yes, your understanding of the Table 1 in the rebuttal is correct. However, these results are for the case where rebalancing is done more frequently than the results reported in Table 1 of the main paper, i.e. every epoch instead of every 10 epochs (as in the main paper). We mention this in the caption of Table 1 in the rebuttal. We apologize for not being more evident about it earlier and thank you for requesting further clarification. We mention at the end of the paragraph above Table 1 in the rebuttal that 'rebalancing only in combination with gradient clipping without using gradient penalty results in $59.62\pm2.21$ mean test accuracy, which is very similar to the bottom row of Table 1 in the rebuttal. As we report in Figure 1 of the PDF attached to the global rebuttal, rebalancing more frequently (such as every epoch) is generally better for a wider range of learning rates, which is why this experiment was also conducted similarly.
> >
> > To make these results with and without gradient penalty comparable to Table 1 in the paper, we redo the experiment by rebalancing every 10 epochs. The results are reported in Table 2 below. Due to the time constraint of the remaining discussion period, the following results are obtained with 5000 epochs of training, instead of 10000 epochs used for the results in Table 1 in the main paper. Thus, it is not directly comparable. Therefore, we also run the experiment for the case without gradient regularization and with gradient clipping using 5000 epochs, which results in a mean test accuracy of $53.6\pm2.37$, which is higher than that obtained with gradient regularization (both with and without gradient clipping) as shown in Table 2 below.
> >
> > Table 2: Mean test accuracy (%) over 10 runs for \texttt{roman-empire} dataset on GAT using gradient l2-norm penalty (of strength $\lambda$) with and without gradient clipping (GC) while rebalancing w.r.t relative gradients every 10 epochs during training. Rebalancing only in combination with gradient clipping without using gradient penalty results in a
> >  mean test accuracy of  $53.6\pm2.37$.
> > | $\lambda$  | $0.01$           | $0.1$            | $2$               | $10$              |
> > | :---------------------: | :-----------------: | :-----------------: | :-----------------: | :-----------------: |
> > | w/o GC   | $50.06\pm2.8$  | $49.90\pm2.92$  | $49.37\pm2.98$ | $51.61\pm1.84$ |
> > | with GC  | $52.08\pm3.21$ | $52.45\pm3.19$ | $50.25\pm3.29$ | $52.09\pm3.01$ |
> >
> > Moreover, we would like to emphasize that the main contribution of our work is dynamic rebalancing, particularly using a criterion based on relative gradients. While gradient regularization could aid in stabilizing training dynamics, we find that gradient clipping is more effective for generalization in our context. We hypothesize that gradient regularization could conflict with our balancing criterion of relative gradients. Yet, a deeper study of gradient regularization is beyond the scope of this paper, because our focus here is to study the impact of rescaling on learning dynamics that leads us to several novel insights, as we discuss in the main paper and global rebuttal.

---

> ### Comment · Reviewer_2KpP · 2024-08-13
>
> I thank the authors for their prompt response on my concerns. I especially appreciate them redoing their experiments by performing the rebalancing every 10 epochs, which gives a better idea of the improvement that comes from rebalancing vs the choice of the gradient stabilizer, i.e., gradient clipping (GC) and / or gradient penalty (GP).
>
> However, my concern that the authors do not adequately establish a relationship between their rescale mechanism and the choice of gradient stabilization approach (for which they have found GC to be optimal) is still concerning. The two must be very intimately intertwined since they both directly affect the scales of the gradient updates to the weights, which is then main premise that this work is meant to study. Although the authors' claim that "a deeper study of gradient regularization is beyond the scope of this paper", developing a more complete understanding (beyond it being presented as just a hyperparameter) of how the choice of the gradient stabilization mechanism, be it GC or GP, affects the proposed dynamic rebalancing process, feels imperative.

---

> > ### Author Response · Authors · 2024-08-13
> >
> > To gain more insights regarding the relationship of dynamic rebalancing (DR), gradient clipping (GC), and regularization using gradient l2-norm penalty (GP), we conduct further experiments and consolidate all results in the following Table 3 to present a systematic ablation study.
> >
> > Table 3: Ablation study between dynamic rebalancing (DR), gradient clipping (GC), and gradient l2-norm penalty (GP) with regularization strength $\lambda$.
> > | DR | GC | GP ($\lambda$) | Avg. Test Acc. $(\%) \pm 95\%$ CI |
> > | :-------------------- | :---------------- | :-------------------------- | :------------------------ |
> > | None                  | No                | No                          | $\mathbf{46.94 \pm 3.26}$   |
> > |                       |                   | Yes (0.01)                 | $45.93 \pm 4.27$       |
> > |                       |                   | Yes (0.1)                  | $44.66 \pm 4.19$       |
> > |                       |                   | Yes (2)                     | $45.09 \pm 3.05$       |
> > |                       |                   | Yes (10)                    | $43.91 \pm 4.05$       |
> > |                       | Yes               | No                          | $36.1 \pm 2.96$        |
> > | every $10$ epochs            | No                | No                          | $49.48 \pm 3.4$        |
> > |                       |                   | Yes (0.01)                 | $50.06 \pm 2.8$        |
> > |                       |                   | Yes (0.1)                  | $49.9 \pm 2.92$        |
> > |                       |                   | Yes (2)                     | $49.37 \pm 2.98$       |
> > |                       |                   | Yes (10)                    | $51.61 \pm 1.84$       |
> > |                       | Yes               | No                          | $\mathbf{54.22 \pm 2.34}$   |
> > |                       |                   | Yes (0.01)                 | $52.08 \pm 3.21$       |
> > |                       |                   | Yes (0.1)                  | $52.45 \pm 3.19$       |
> > |                       |                   | Yes (2)                     | $50.25 \pm 3.29$       |
> > |                       |                   | Yes (10)                    | $52.09 \pm 3.01$       |
> > | every epoch                 | No                | No                          | $51.32 \pm 3.35$       |
> > |                       |                   | Yes (0.01)                 | $50.43 \pm 3.07$       |
> > |                       |                   | Yes (0.1)                  | $52.76 \pm 2.75$       |
> > |                       |                   | Yes (2)                     | $51.4 \pm 3.42$        |
> > |                       |                   | Yes (10)                    | $52.44 \pm 2.43$       |
> > |                       | Yes               | No                          | $59.62 \pm 2.21$       |
> > |                       |                   | Yes (0.01)                 | $59.28 \pm 2.55$       |
> > |                       |                   | Yes (0.1)                  | $\mathbf{59.82 \pm 2.64}$   |
> > |                       |                   | Yes (2)                     | $58.4 \pm 1.85$        |
> > |                       |                   | Yes (10)                    | $59.7 \pm 2.35$        |
> >
> > We find that the reviewer's original intuition on GP being a better alternative to GC is indeed correct under regular training as we compare cases of training without DR but with a) either GP or GC and b) neither GP nor GC, where we find the results for the case with and without GP to be quite similar and thus with further tuning, may also potentially improve performance. However, in the case of training with DR, the use of GC seems to be more critical than GP. We hypothesize that GP (that takes place in a continuous way and affects possibly every gradient in every epoch) may interfere with the rebalancing w.r.t. relative gradients more than GC (which is discrete, more likely affects fewer gradients more rarely and can better handle rare cases of exploding gradients that may arise due to any numerical instability caused by direct manipulation of gradients during DR). Nevertheless, in combination with GC, GP at least doesn't seem to harm the performance and even further improves the performance slightly in the case of more frequent rebalancing (although the difference in this case of using both GC and GP and using only GC seems to be very small). Therefore, we conclude that frequent DR with both GC and GP seems to be the optimal case.
> >
> > We hope this answers the reviewer's question and thank them for their valuable feedback which has led us to further insights regarding training under different conditions.

---

> > > ### Comment · Reviewer_2KpP · 2024-08-13
> > >
> > > I thank the authors for their diligence. The above experiment, and equally importantly, the accompanying explanation describing the possible interplay between the proposed dynamic rescaling and the associated choice of gradient stabilizer addresses my main concerns. With this, based on my considerations, the authors reasonably study the core aspects directly associated with their proposed approach, which I believe qualifies this work for acceptance. I am thus increasing my score.
> > >
> > > Since the choice of the gradient stabilizer affects the behaviour of their algorithm at a fundamental level, I hope the authors include their latest findings, as well as their explanation detailing the interplay between dynamic rescaling and gradient stabilizers in the main manuscript, should it be accepted.

---

> > > > ### Author Response · Authors · 2024-08-14
> > > >
> > > > We thank the reviewer for their acknowledgment and valuable feedback throughout the rebuttal process. As suggested, we will include our latest findings and insights in the paper.

---

### Author Rebuttal · Authors · 2024-08-07

We thank the reviewers for their encouraging comments and constructive suggestions that we have incorporated to improve the paper. We jointly answer some key common questions of the reviewers here in two parts regarding i) additional insights and ii) limitations.

**Further insights and additional experiments:**

1. We discover trends regarding homophily and heterophily in the training dynamics of GATs.

* *Background:* In the context of GNNs, the optimal performance achieved by a model is, to a large extent, dependent on how well the inductive bias of the model architecture aligns with the task and its underlying graph structure. For example, it is widely known that general GNNs, without specially-introduced architectural elements, such as GCN perform better on homophilic than on heterophilic tasks.

* *Observation:* We find that training in balance is generally more effective for heterophilic tasks (as we report in the main paper) than homophilic tasks (such as Cora and Citeseer) where we observe a similar performance with no significant advantage of training in balance. On the contrary, allowing learning focused on the first layer may be beneficial for homophilic tasks, although the effect is not similar in magnitude to that of balancedness for heterophilic tasks. For example, see the test accuracy reported for different influences on gradient dynamics in Fig. 3.

* *Explanation:* Intuitively, we hypothesize that homophilic tasks rely more on the neighborhood aggregation functionality of GNNs rather than feature learning. In this case, an aggregation over a random transformation of similar features may still be sufficient for good generalization. This is in line with a recent analysis of training dynamics [1] which shows that on homophilic graphs, alignment of the underlying structure with the optimal kernel matrix allows parameter-free methods similar to label propagation to perform at par with GNNs. However, this adversely affects generalization on heterophilic tasks where the graph structure does not align with the optimal kernel. In other words, the neighborhood is not very relevant for a node's label in heterophilic settings and thus the node relies more on feature learning rather than neighborhood aggregation. This is also supported by results showing that embedding additional MLP layers in the network significantly improves the performance of basic GNNs such as GATs on these heterophilic tasks [2]. Thus, we conclude that training in balance to learn better feature transformations in all layers (and potentially neighbors farther away in deeper models) is more effective in heterophilic cases. From the perspective of neighborhood aggregation,  dynamic rescaling allows the model to achieve higher values of $\alpha_{ii}$ (i.e. the weight assigned to the node itself in GATs, see Fig. 3) by potentially allowing better training of the attention parameters whereas an opposite but not equally large effect can be observed for homophilic settings with a larger focus on the neighborhood (and thus smaller $\alpha_{ii}$) may be better. For sufficiently shallow models, however, the difference in regular and layer-focused training could be negligible for homophilic tasks.

2. We induce grokking-related behavior on a real-world dataset (see Fig. 4) in two steps. Firstly, we allow only the last (or second to last) layer to learn which allows the training accuracy to increase continually while the test accuracy saturates or begins to drop. At this point, we rescale the network to bring all layers in balance w.r.t. relative gradients, following which,  the test accuracy immediately begins to improve more rapidly accompanied by a drop in training accuracy. This can be interpreted as the network `learning' more effectively rather than overfitting to the training data. While this is different from grokking where the training accuracy would generally not drop, it is independently an interesting observation on a real-world dataset.

3. We show that rebalancing the network during training by dynamic rescaling further improves the generalization brought about by larger learning rates (see Fig. 1).

**Limitations:** As suggested by multiple reviewers, we discuss the limitations elaborately in a dedicated section that we will include in the paper as follows.

1. The primary limitation of dynamically rescaling a model is that we require the rescale invariance of the model architecture (if it exhibits one), which may vary widely across different GNN architectures. The dynamic rescaling proposed in the paper applies to GCNs and GATs, two standard baseline GNN models commonly used to study and develop insights into the training dynamics of GNNs.

2. From an implementation perspective, directly manipulating model parameter values and gradients during rescaling can result in numerical instability issues that we currently regulate using gradient clipping, a commonly used practice in machine learning. Nevertheless, a more principled approach to tackling this problem could be beneficial.

3. Rebalancing repeatedly and frequently during training may incur computational overhead in practice. However, it may be offset by the increased training speed, requiring fewer epochs overall. From a time complexity perspective, rebalancing only adds a constant factor of operations in each epoch, determined by the number of iterations in one rebalancing step. In practice, we find that only a few iterations (<10) are necessary to balance the network approximately enough to gain better generalization and/or training speed.

[1] Yang et al. How Graph Neural Networks Learn: Lessons from Training Dynamics. (2024).

[2] Platonov et al. A critical look at the evaluation of GNNs under heterophily: Are we really making progress? (2023).

---

### Decision · Program_Chairs · 2024-09-25

**Decision:**

Accept (poster)

**Comment:**

This work studies dynamic rescaling during GNN training to improve learning speed and generalization. The reviewers overall appreciated the novel use of relative gradients and interesting experiments conducted, along with widely acknowledged useful and relevant empirical results, though with some concerns about significance. There was a robust discussion during rebuttal period and additional clarifications and content to overcome some weaknesses should be made by the authors for a revised manuscript.

Additional context after discussion with the SAC: Reviewer TkPU wanted more effective clarification of the motivation and details on learning rate and grokking. The authors included the additional detail in two follow up responses to address their concerns and improved the paper. Reviewer 2KpP following a discussion, was satisfied with the rebuttal assuming the authors included data from the discussion in the final paper. Overall, while this was borderline, it fell on the side of acceptance.